# Hypertonicity-Affected Genes Are Differentially Expressed in Clear Cell Renal Cell Carcinoma and Correlate with Cancer-Specific Survival

**DOI:** 10.3390/cancers12010006

**Published:** 2019-12-18

**Authors:** Siarhei Kandabarau, Janna Leiz, Knut Krohn, Stefan Winter, Jens Bedke, Matthias Schwab, Elke Schaeffeler, Bayram Edemir

**Affiliations:** 1Dr. Margarete Fischer-Bosch Institute of Clinical Pharmacology, 70376 Stuttgart, Germany; Siarhei.Kandabarau@ikp-stuttgart.de (S.K.); Stefan.Winter@ikp-stuttgart.de (S.W.); matthias.schwab@ikp-stuttgart.de (M.S.); elke.schaeffeler@ikp-stuttgart.de (E.S.); 2University of Tübingen, 72074 Tübingen, Germany; 3Department of Medicine, Hematology and Oncology, Martin Luther University Halle-Wittenberg, 06120 Halle (Saale), Germany; 4Max Delbrück Center for Molecular Medicine (MDC), 13125 Berlin, Germany; 5Core Unit DNA–Technologien, Medizinische Fakultät, Universität Leipzig, 04103 Leipzig, Germany; krok@medizin.uni-leipzig.de; 6Department of Urology, University Hospital Tübingen, 72076 Tübingen, Germany; jens.bedke@med.uni-tuebingen.de; 7German Cancer Consortium (DKTK) and German Cancer Research Center (DKFZ), 69120 Heidelberg, Germany; 8Departments of Clinical Pharmacology, Pharmacy and Biochemistry, University Tübingen, 72076 Tübingen, Germany; 9iFIT Cluster of Excellence EXC 2180, University of Tübingen, 72076 Tübingen, Germany

**Keywords:** gene signature, renal cancer, survival prediction

## Abstract

The heterogeneity of renal cell carcinoma (RCC) subtypes reflects the cell type of origin in the nephron, with consequences for therapy and prognosis. The transcriptional cues that determine segment-specific gene expression patterns are poorly understood. We recently showed that hypertonicity in the renal medulla regulates nephron-specific gene expression. Here, we analyzed a set of 223 genes, which were identified in the present study by RNA-Seq to be differentially expressed by hypertonicity, for the prediction of cancer-specific survival (CSS). Cluster analyses of these genes showed discrimination between tumor and non-tumor samples of clear cell RCC (ccRCC). Refinement of this gene signature to a four-gene score (OSM score) through statistical analyses enabled prediction of CSS in ccRCC patients of The Cancer Genome Atlas (TCGA) (*n* = 436) in univariate (HR = 4.1; 95% CI: 2.78−6.07; *p* = 4.39 × 10^−13^), and multivariate analyses including primary tumor (T); regional lymph node (N); distant metastasis (M); grading (G)(*p* = 2.3 × 10^−5^). The OSM score could be validated in an independent ccRCC study (*n* = 52) in univariate (HR = 1.29; 95% CI = 1.05–1.59; *p* = 0.011) and multivariate analyses (*p* = 0.016). Cell culture experiments using RCC cell lines demonstrated that the expression of the tumor suppressor *ELF5* could be restored by hypertonicity. The innovation of our novel gene signature is that these genes are physiologically regulated only by hypertonicity, thereby providing the possibility to be targeted for therapy.

## 1. Introduction

The kidney’s anatomy and histology consists of different renal cell types located at defined parts of the kidneys, reflected in the complexity of renal function. This is also reflected by the heterogeneity of renal cell carcinoma (RCC) subtypes [1]. The main subtypes are clear cell (ccRCC), papillary (pRCC) and chromophobe (chRCC) renal cell carcinoma [2]. Although several targeted therapies are currently applied, survival rates—especially for metastatic RCC—are still low and innovative treatment strategies are needed [2].

Comprehensive studies carried out by The Cancer Genome Atlas (TCGA) provided further insight into the evolution and origin of RCC, for example, by identifying gene signatures that enable discrimination between the RCC subtypes or define the cell type of origin. Moreover, it was found to be possible to predict clinical outcome in ccRCC patients based on gene expression similarity to the proximal tubule of the nephron, which is the presumed origin of ccRCC [3]. Recently, another study analyzed the impact of different gene expression profiles on RCC ontogeny [1]. The authors were able to identify gene expression programs that were specific for a distinct nephron segment and were also present in the corresponding RCC subtypes. Both studies used data based on nephron-specific gene expression patterns and were able either to improve the prediction of patient outcome or identify gene expression networks defining the origin of RCC. However, as mentioned by Lindgren et al. [1], the transcriptional cues that determine segment-specific gene expression patterns are only partly understood. We have recently shown that the unique hypertonicity in the renal inner medulla regulates kidney and nephron-specific gene expression [4]. The group of Prof. Ian Frew showed that deletion of renal expression of the tumor suppressor von Hippel–Lindau (VHL) protein altered the urine concentration capability in mice [5]. They postulate that the mice cannot build up the hyperosmotic gradient in the kidneys that is necessary for urine concentration. The transcription factor nuclear factor of activated T-cells 5 (*NFAT5*) is the main transcription factor activated by the hyperosmotic environment, and induces the expression of several genes [6]. A recent study in that Special Issue of *Cancers* showed that microRNAs that mediate metabolic reprogramming in ccRCC also target *NFAT5* [7]. This was also associated with a reduced level of *NFAT5* target genes in the ccRCC samples compared to solid normal tissue.

In the present study, we analyzed if the hypertonicity-affected genes were also differentially expressed in ccRCC tumor samples and normal tissue, and if these genes were associated with the clinical outcome of the patients.

## 2. Results and Discussion

In contrast to our initial study, where we used microarrays, here we performed RNA-Seq using primary cultured inner medullary collecting duct (IMCD) cells cultivated at 300 or 900 mosmol/kg to identify differentially expressed transcripts affected by hypertonicity (for details see Appendix A). We detected significant differences between the two conditions for 355 transcripts (false discovery rate FDR < 0.05; log_2_ fold change (FC) >3/<−3) and there were matching human transcripts for 284 of these (223 genes) (Figure 1A and Appendix A). Hierarchical clustering of the TCGA Kidney Clear Cell Carcinoma (KIRC) samples based RNA-Seq data using the top 223 hypertonicity-affected genes clearly separated the normal non-tumor tissue samples (*n* = 67) from the tumor samples (*n* = 449; Figure 1B).

Part of the genes (41) showed a log_2_ fold change of >3/<−3 between normal and tumor samples. Interestingly, several of the transcripts induced by hypertonicity were suppressed, and transcripts suppressed by hypertonicity were induced in the tumor samples compared to normal samples (Appendix A).

The effect of hyper-osmolality on gene expression can be reverted by hypo osmotic switch [4]. For example one of the hypertonicity-induced transcripts (0 fragments per kilobase of transcript per million mapped reads (FPKM) at 300 vs. 75 FPKM at 900 mosmol/kg, see Appendix A) was the E74-like ETS transcription factor 5 (*ELF5*). *ELF5* has been described as a tumor suppressor in RCC and is more or less absent in tumor samples (Appendix A) [8]. Since *ELF5* has an important role as a tumor suppressor, we next asked whether it is possible to induce its expression in a ccRCC cell line by hyperosmolality. To test this, we used the established ccRCC cell line 786-0, and the same cell line that ectopically expresses WT-VHL (786-0-VHL). Both were cultivated either under isotonic (300 mosmol/kg) or for different periods of time under hyperosmotic (600 mosmol/kg) conditions. Indeed, the expression of *ELF5* could be induced by cultivation of 786-0 cells under hyperosmotic conditions as shown by PCR or qPCR analyses (Figure 2A,B). Interestingly, the induction of *ELF5* expression was higher in VHL+ cells than in VHL-deficient cells.

Our results clearly indicate that it is possible to induce the expression of the tumor suppressor ELF5 in RCC cells only by osmolality without any genetic manipulation. With the hyper-osmolality, we have identified a pathway that could be targeted for future intervention.

In the next step, we analyzed the predictive value of the hypertonicity-related genes for clinical outcome in ccRCC patients using the Cox proportional hazards model. Out of the 223 genes, 111 (49.8%) showed a significant effect (Table 1).

Within the genes that had a significant impact on patients’ survival, hypertonicity-downregulated genes tend to have a negative effect on survival (32 out of 41) while hypertonicity-upregulated genes have equal number of negative (35) and positive (35) effects. The corresponding data with the gene IDs and fold changes are provided in Appendix A. This data suggests that the expression of hypertonicity-affected genes can be used to predict cancer-specific survival.

We next selected a minimum set of genes necessary for survival prediction using RNA-Seq data from the TCGA KIRC cohort. We identified 4 (*COL1A1*, *NDUFA4L2*, *S100A6*, *MT2A*) out of the 223 different genes that were regulated by hypertonicity in rats and subsequently defined our OSM score based on these four genes (Appendix A). Interestingly, all four genes have previously been associated with ccRCC tumorigenesis [9,10,11,12].

Our novel established OSM score based on these four genes was significantly associated with cancer-specific survival (Figure 3A; HR = 4.1; 95% CI: 2.78–6.07; *p* = 4.39 × 10^−13^; Cox proportional hazards regression model) in the TCGA cohort.

Multivariate analysis of the score together with clinicopathological parameters (T (primary tumor), N (regional lymph node), M (distant metastasis present at diagnosis), G (grading)) indicated that the score significantly predicted cancer-specific survival (*p* = 2.3 × 10^−5^, Table 2).

The independent role for prediction in the multivariate model was proven by analysis of deviance (*p* = 1.2 × 10^−4^). To validate these results, the OSM score was calculated in an independent cohort of ccRCC patients (*n* = 52; for details see [13]) based on expression levels of the selected genes and their model coefficients. Notably, we showed that the OSM score was also significantly associated with cancer-specific survival (Figure 3B; HR = 1.29; 95% CI: 1.05–1.59; *p* = 0.011). Multivariate analysis confirmed its role in the prediction of cancer-specific survival (*p* = 0.016) in our validation cohort (Table 1; analysis of deviance *p* = 0.0215).

A link between loss of *VHL* and osmolality has also been shown using kidney-specific VHL knock-out mice [5]. The authors observed that the mice had increased diuresis. The same group developed a renal cancer mice model and investigated the gene expression profile in mouse ccRCCs and kidney cortices [14]. Using these gene expression data, we could demonstrate that the mouse ccRCCs and normal kidney cortices could be discriminated based on the osmolality-regulated genes (Appendix A). Our results indicate that *VHL* function is important for hyper-osmolality-induced gene expression, as seen for *ELF5*. In a recent manuscript that we have submitted to *Cancers* we were able to show that the deletion of *VHL* also reduced the expression of several other hyper-osmolality-induced genes. This implies that *VHL* is prominently involved in the regulation hyper-osmolality-induced pathways. Since up to 85% of RCC patients harbor loss of *VHL* function, it is mandatory to identify the underlying cellular and molecular mechanisms.

In summary, our in vitro and in vivo data demonstrate that osmolality is an interesting pathway for the future development of drugs or other interventions in ccRCC which has not been considered so far. Moreover, this is the first report that defines an expression pattern of genes that can not only be used to discriminate between normal vs. tumor tissue and is associated with cancer-specific survival in independent ccRCC cohorts, but have a common physiological mechanism regulating their expression. Thus, targeting osmolality represents a novel interesting option for ccRCC therapy development, and further studies are warranted to identify the functional relevance of hypertonicity-associated pathways in tumor development and proliferation.

## 3. Materials and Methods

### 3.1. Primary Renal Cell Culture and RNA-Seq

Experiments were approved by a governmental committee on animal welfare (Landesamt für Natur, Umwelt und Verbraucherschutz Nordrhein-Westfalen, Germany) and were performed in accordance with national animal protection guidelines (A 60/1993 and A 67/09).

Primary cultured IMCD cells were prepared as described before [4]. For each group, three biological replicates were used. The groups included cells which had been cultivated at 300 or 900 mosmol/kg for one week. Total RNA was isolated using the mirVana miRNA Isolation Kit (Thermo Scientific, Waltham, MA, USA); 500 ng of total RNA were depleted of ribosomal RNA using the RiboMinus kit (Thermo Fisher Scientific, Waltham, MA, USA) according to the manufacturer’s instructions. Purified RNA was then fragmented by the addition of fragmentation buffer (200 mM Tris acetate, pH 8.2, 500 mM potassium acetate, and 150 mM magnesium acetate) and heating at 94 °C for 3 min in a thermocycler followed by ethanol precipitation with ammonium acetate and GlycoBlue (Thermo Fisher Scientific) as carrier. Fragmented RNA was then reverse transcribed using random hexamer and Superscript III (Thermo Fisher Scientific). The second strand was synthesized using the TargetAmp kit (Epicentre, Madison, WI, USA) according to the manufacturer’s instructions. The final steps of library preparation (e.g., blunt end repair, adapter ligation, adapter fill-in, and amplification) were done according to Meyer and Kircher [15]. The barcoded libraries were purified and quantified using the Library Quantification Kit (Illumina/Universal; KAPA Biosystems, Wilmington, MA, USA) according to the manufacturer’s instructions. A pool of up to 10 libraries was used for cluster generation at a concentration of 10 nM using an Illumina cBot. Sequencing of 2 × 100 bp was performed with an Illumina HiScanSQ sequencer at the sequencing core facility of the IZKF Leipzig (Faculty of Medicine, University Leipzig) using version 3 chemistry and flowcell according to the instructions of the manufacturer. Demultiplexing of raw reads, adapter trimming, and quality filtering were done according to Stokowy et al. [16] using TruSeq (Illumina) adapter sequences.

### 3.2. 786-0 Renal Cancer Cell Line and Real-Time PCR

The 786-0 and VHL-expressing 786-0-VHL were a kind gift of Prof. Barbara Seliger and were cultivated as described in [17]. For experimental setting, the cell culture medium was adjusted to 600 mosmol/kg by the addition of 100 mM NaCl and 100 mM urea. The cells were cultivated for different time points at 600 mosmol/kg. Total RNA isolation and cDNA synthesis were performed as described previously [4]. Real-time PCR was performed using the SYBR Green PCR Master Mix with the ABI PRISM 7900 Sequence Detection System. All instruments and reagents were purchased from Applied Biosystems (Darmstadt, Germany). Relative gene expression values were evaluated with the 2^−ΔΔCt^ method using *GAPDH* as reference gene [18]. The primer sequences for *ELF5* are *ELF5*-sense CGT GGA CTG ATC TGT TCA GCA ATG A, *ELF5*-antisense CAG GGT GGA CTG ATG TCC AGT ATG A and for *GAPDH GAPDH*-sense CAA GCT CAT TTC CTG GTA TGA C and *GAPDH*-antisense GTG TGG TGG GGG ACT GAG TGT GG.

### 3.3. Study Cohorts

Publicly available gene expression data from The Cancer Genome Atlas (TCGA) from a cohort of ccRCC patients (KIRC cohort, *n* = 449) were used to compare osmolality-induced genes expression between tumor and non-tumor samples. In this data set, 436 patients had both expression and CSS data and were used to develop the osmolality score. Expression data from tumor and non-tumor tissue were downloaded using the Bioconductor TCGAbiolinks package.

The validation cohort consisted of primary tumors with ccRCC histology (*n* = 52) of patients treated at the Department of Urology, University Hospital Tuebingen, Germany. Details of the study and tissue sample collection were described previously [13]. Transcriptome analyses was performed using the Human Transcriptome Array HTA 2.0 (Affymetrix/Thermo Fisher Scientific, Waltham, MA, USA), as described previously by Büttner et al. [13]. The accession number for genome-wide data at the European Genome-phenome Archive (EGA) (www.ebi.ac.uk/ega/home), which is hosted by the EBI and the CRG, is EGAS00001001176. Cancer-specific survival was used as endpoint in survival analyses of the development cohort (ccRCC KIRC) and the validation cohort, as described previously [13].

### 3.4. Statistical Analyses

RNA-Seq reads were aligned using bowtie2 and tophat2 to the rat reference genome (rnor6) according to Kim et al. [19]. Rat mRNA-Seq read counts were normalized and tested for differential expression using the Bioconductor edgeR package (v 3.24.3, [20]); 355 rat transcripts showed significant difference between two conditions: 900 and 300 mosmol/kg (Benjamini–Hochberg [21] adjusted *p*-Value < 0.05; log_2_FC >3/<−3). Of these, 284 of them had matching human transcripts by gene symbol. For those 284 transcripts (223 genes), mRNA-Seq expression values (FPKM-UQ) of the TCGA-KIRC cohort (449 tumor samples with 67 matching tissue normal samples) were clustered (hierarchical clustering with agglomeration method ward. D2 and Euclidean distance). Clustering proved that selected transcripts expression clearly discriminated between tumor and normal samples. The potential impact of those genes’ expression on TCGA-KIRC patients’ (*n* = 436) cancer-specific survival (CSS) was tested by building a Cox proportional hazards model on each gene’s expression separately (survival R package v 3.1-7, [21]). We found that 111 genes showed significant effect (Benjamini–Hochberg adjusted *p*-Value < 0.05). Later, we built a Cox proportional hazards model with lasso penalty based on the expression of the entire set of 223 genes (glmnet R package v 2.0-16, [22]). Four genes had non-zero coefficients according to the model, with minimal cross-validation error. Each TCGA-KIRC patient was assigned a survival score (termed the OSM score) calculated as the weighted sum of the expression of the four selected genes multiplied by the respective model coefficient. Analogously, the score was calculated for our independent RCC cohort (*n* = 52; [13]) based on selected genes expression (determined by microarray analyses) and the respective model coefficients. The value was multiplied by 100,000 times to avoid infinite hazards ratio. Patient cohorts were recursively partitioned based on the survival score using conditional inference trees [23] with the endpoint CSS. Multivariate survival analysis was performed using Cox proportional hazards regression models. Comparison of Cox models (with and without OSM score) was done using analysis of deviance [24].

## 4. Conclusions

Our study demonstrates that osmolality is an interesting pathway in ccRCC which has not yet been considered. The expression of hypertonicity-regulated genes is clearly associated with cancer-specific survival in ccRCC. We were also able to induce the expression of potentially tumor-suppressive genes by cultivating ccRCC cell lines under hyper-osmotic conditions.

Thus, targeting osmolality-associated pathways might represent a novel interesting therapeutic option for ccRCC.

## Figures and Tables

**Figure 1 cancers-12-00006-f001:**
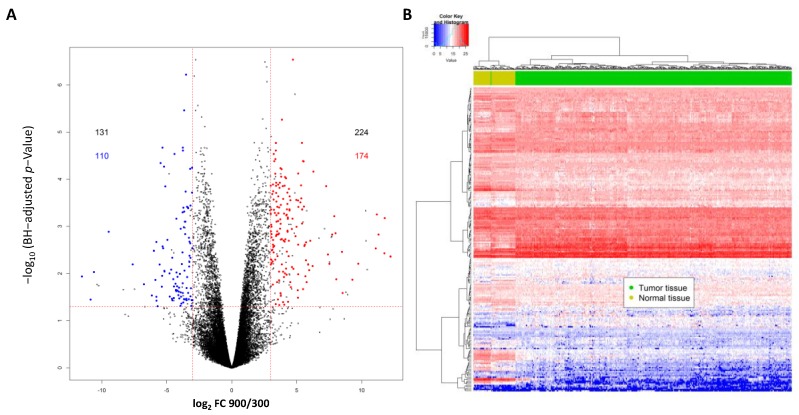
(**A**) Differentially expressed transcripts affected by hypertonicity in primary cultured inner medullary collecting duct (IMCD) cells either cultivated at 300 or 900 mosmol/kg. In total, 355 transcripts were differentially expressed (with a cut off log_2_ fold change of >3 and <−3) and there were matched human transcripts for 284 of those. Of those, 110 transcripts were downregulated and 174 transcripts were upregulated by hypertonicity. (**B**) Hierarchical clustering of samples from The Cancer Genome Atlas (TCGA) Kidney Clear Cell Carcinoma (KIRC) cohort based on the hypertonicity-affected genes. The expression levels of the top 223 regulated genes were extracted from the TCGA KIRC cohort, and hierarchical clustering was performed. This set of genes was able to clearly separate clear cell renal cell carcinoma (ccRCC) samples (dark green) from the normal tissue samples (light green).

**Figure 2 cancers-12-00006-f002:**
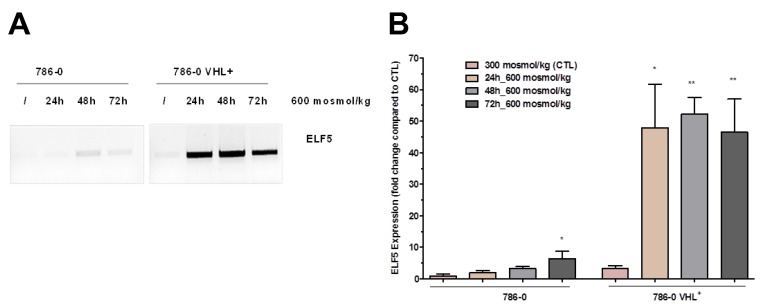
E74-like ETS transcription factor 5 (*ELF5*) expression in 786-0 cells is induced by environmental hypertonicity. von Hippel–Lindau (VHL)-deficient 786-0 and VHL-expressing (VHL+) 786-0 cells were cultivated either in normal medium (300) or for 24, 48, and 72 h in 600 mosmol/kg medium. (**A**) The expression of *ELF5* was analyzed by PCR. The osmolality was increased by the addition of 100 mM NaCl and 100 mM urea. The expression of *ELF5* was *VHL* dependent. (**B**) The expression of *ELF5* was quantified by real-time PCR in VHL-deficient 786-0 and VHL-expressing 786-0 VHL+ cells (*N* > 3, *p*-Value < 0.05 compared to control (CTL) using one-way ANOVA are marked by *, *p*-Value < 0.01 are marked by **). For more details about the PCR product of ELF5, please view the Appendix A.

**Figure 3 cancers-12-00006-f003:**
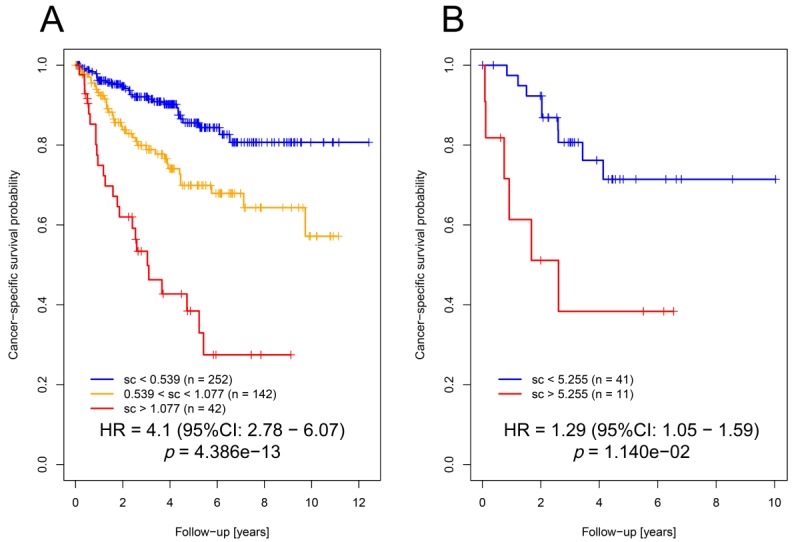
The OSM score could predict patient survival. (**A**) Kaplan–Meier plot indicating that hypertonicity-affected genes using the four selected genes (OSM score) can be applied for the prediction of patients’ cancer-specific survival in the TCGA KIRC cohort. (**B**) Kaplan–Meier plot indicating that the OSM score can be applied for the prediction of patients’ cancer-specific survival in the ccRCC validation cohort.

**Table 1 cancers-12-00006-t001:** Number of hypertonicity-affected genes and their impact on patient survival.

Effect on Cancer Specific Survival	Downregulated	Upregulated
hazardous	32	35
indifferent	46	66
favorable	9	35

**Table 2 cancers-12-00006-t002:** Multivariate Cox regression for cancer-specific survival in the TCGA cohort (*n* = 409) and the validation cohort (*n* = 51).

Multivariate Analyses	Variable	Level	*p*-Value (Wald Test)	HR (95% CI)
**Including T, N, M, G and OSM score**	**OSM score**		2.35 × 10^−5^	2.6 (1.67–3.69)
**(TCGA cohort)**	**Primary tumor**	T1/T2		1
		T3/T4	5.05 × 10^−2^	1.7 (1–2.86)
	**Lymph nodes**	N0		1
		N1	6.15 × 10^−2^	2.71 (0.95–7.72)
		NX	2.4 × 10^−2^	0.6 (0.38–0.93)
	**Distant metastasis**	M0		1
		M1	1.55 × 10^−12^	5.62 (3.48–9.07)
	**Grade**	G1/G2		1
		G3/G4	6.69 × 10^−3^	2.35 (1.27–4.37)
**Including T, N, M, G and OSM score**	**OSM score**		1.58 × 10^−2^	1.34 (1.06–1.7)
**(Validation cohort)**	**Primary tumor**	T1/T2		1
		T3/T4	2.54 × 10^−1^	2.63 (0.5–13.9)
	**Lymph nodes**	N0		1
		N1/N2	1.27 × 10^−1^	0.33 (0.08–1.36)
	**Distant metastasis**	M0		1
		M1	2.78 × 10^−5^	41 (7.22–233.06)
	**Fuhrman grade**	G1/G2		1
		G3/G4	9.94 × 10^−1^	1 (0.29–3.43)

Abbreviations: CI, confidence interval; HR, hazard ratio; Ref., reference level; T, primary tumor; N, regional lymph node; M, distant metastasis present at diagnosis; G, grading. Cases with grading information “GX” or metastasis status “MX” were excluded from multivariate analyses. OSM scores were determined based on gene expression data measured by RNA-Seq (TCGA) or microarray analyses (validation cohort).

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
