# Peer review of "Hypertonicity-Affected Genes Are Differentially Expressed in Clear Cell Renal Cell Carcinoma and Correlate with Cancer-Specific Survival"

_cancers, 2019, doi:10.3390/cancers12010006_

Round 1

Reviewer 1 Report

The manuscript entitled “Hypertonicity affected genes are differentially expressed in clear cell renal cell carcinoma with consequences for cancer-specific survival” addresses the effects of hypertonicity on gene expression and survival in clear cell renal carcinoma patients. The authors identified hypertonicity associated genes in a rat model and tested the ability of human orthologs to distinguish normal from neoplastic kidney cells and to prognosticate patient survival. This is an interesting and appropriately designed study with, however, several flaws.

There is no mention of authorization for animal experiments.

The statistical analysis of RNAseq data is apparently based on simple vulcano plots that take into account p-values and fold changes. More advances class comparisons such as bootstrapping test or linear modeling should be applied.

It is not clear why 223 genes were selected (likely based on the possibility to identify human orthologs).

The genes selected identify two clusters yet the gene expression differences appear to be marginal. This could be due to the imperfect feature selection procedure (see above).

The authors also address the functional implication of one of these genes, ELF5. One would be interested to know the effects of overexpression of ELF5 on the hypertonicity signature in the cellular model used. Gene enrichment analysis should be performed on the differentially expressed genes.

Figure 3 is not informative and should be substituted by a table showing the actual data

Figure 4 (low quality, legends within the figure are hardly readable). The subdivision in prognostic groups appears arbitrarily. In the absence of specific indications only two groups based on the median values should be used for KM curves. The survival analysis should in addition be performed by mixed models containing both, existing prognostic factors and the gene score in order to evaluate eventual clinical significance of the score.

Throughout the manuscript there are incomplete or incorrect sentences that need rephrasing.

Reviewer 2 Report

The authors have made a novel and important discovery.  Building on their previous experience demonstrating that hypertonicity regulates nephron specific gene expression they have identified a 4-gene panel (OSM score) that is tightly correlated with cancer-specific survival in ccRCC.  The methods and analyses are well described and appropriate for the questions being asked.  They have confirmed the results in independent cohorts and have demonstrated that the expression of a clinically relevant gene (ELF5) can be induced with hypertonicity in cultured human RCC cells.  The idea of tonicity controlling renal cell's gene expression is elegant, novel and consonant with known physiology.  The demonstration that the expression of  this family of genes is relevant to clinical RCC is important and novel.

Despite the novelty and importance of the findings, the authors have consistently overstated the conclusions.  They have shown correlation not causation.  This overstatement needs to be corrected:

1.  The title is misleading because the words "with consequences for" implies causality.  Suggest deleting these three word and replacing with "and correlate with."

2.  Page 4, lines 11 and 12.  the sentence "Thus targeting osmolality affected pathways represent a novel promising therapeutic option for ccRCC" is an overstatement.  This should be changed to be similar to the statement in the abstract, indicating that they have identified a pathway that could be targeted for future intervention.  There is no "therapeutic option" at this point in time.

3.  Page 6 lines 23-26.  This is the same problem as in #2 above.  This is not a "therapeutic option" but an interesting pathway for the future development of drugs or other interventions.

4.  Page 8, line 12.  Same as #2 and #3 above.

A second concern is that the cells used including 786-O and 786-O VHL+ have not been genotyped confirmed to be the cells that they claim them to be.  This is a current standard for publication.

A third concern is that the effect of hypertonicity on ELF5 expression appears to depend on the expression of VHL (Figure 2).  Up to 85 % of clinical RCC has either downregulated or completely eliminated the expression of VHL.  Does this limit the future potential therapeutic targeting to the 15-20% of RCC cases with preserved VHL expression?  This needs to be addressed in a short paragraph in the discussion.

Reviewer 3 Report

The manuscript entitled "Hypertonicity affected genes are differentially expressed in clear cell renal cell carcinoma with consequences for cancer-specific survival " by Kandabarau et al inherited about hypertonicity, for prediction of cancer-specific survival (CSS) in renal cancer cell lines requires major revisions for pubblication:

why these type ov values  (300 or 900 31 mosmol/kg ) was adopted in this experiment? could the authors report their suggesions? could the author evaluate if ELF5 differential expression was related to hypertonicity by approaching control study with clonal expression of deleted domani protein? Please, could the authors explain why statistical analysis were only        performed on  RCC cell lines and other renal cell lines were excluded? please, could the authors define if thys study may affect other renal cell lines?

Round 2

Reviewer 1 Report

The authors have adequately addressed the issues raised. Several typos need to be corrected.

Author Response

We thank the reviewer for the helpful comments and we are happy that we could address all the  issues.

We have also revised the manuscript for English language and typos.

Beste regards,

Bayram Edemir

Reviewer 3 Report

The manuscript is suitable for pubblication without any revision.

Author Response

We thank the reviewer for the helpful comments and we are happy that we could address all the  issues.

Best regards,

Bayram Edemir